# Shape-Memory Polymeric Artificial Muscles: Mechanisms, Applications and Challenges

**DOI:** 10.3390/molecules25184246

**Published:** 2020-09-16

**Authors:** Yujie Chen, Chi Chen, Hafeez Ur Rehman, Xu Zheng, Hua Li, Hezhou Liu, Mikael S. Hedenqvist

**Affiliations:** 1State Key Laboratory of Metal Matrix Composites, School of Materials Science and Engineering, Shanghai Jiao Tong University, Shanghai 200240, China; yujiechen@sjtu.edu.cn (Y.C.); Chenchi0707@sjtu.edu.cn (C.C.); zhengxu-sun@sjtu.edu.cn (X.Z.); 2Collaborative Innovation Centre for Advanced Ship and Dee-Sea Exploration, Shanghai Jiao Tong University, Shanghai 200240, China; lih@sjtu.edu.cn (H.L.); hzhliu@sjtu.edu.cn (H.L.); 3Department of Fibre and Polymer Technology, School of Engineering Sciences in Chemistry, Biotechnology and Health, KTH Royal Institute of Technology, SE-100 44 Stockholm, Sweden

**Keywords:** shape-memory, artificial muscle, polymer, liquid crystal elastomer

## Abstract

Shape-memory materials are smart materials that can remember an original shape and return to their unique state from a deformed secondary shape in the presence of an appropriate stimulus. This property allows these materials to be used as shape-memory artificial muscles, which form a subclass of artificial muscles. The shape-memory artificial muscles are fabricated from shape-memory polymers (SMPs) by twist insertion, shape fixation via T_m_ or T_g_, or by liquid crystal elastomers (LCEs). The prepared SMP artificial muscles can be used in a wide range of applications, from biomimetic and soft robotics to actuators, because they can be operated without sophisticated linkage design and can achieve complex final shapes. Recently, significant achievements have been made in fabrication, modelling, and manipulation of SMP-based artificial muscles. This paper presents a review of the recent progress in shape-memory polymer-based artificial muscles. Here we focus on the mechanisms of SMPs, applications of SMPs as artificial muscles, and the challenges they face concerning actuation. While shape-memory behavior has been demonstrated in several stimulated environments, our focus is on thermal-, photo-, and electrical-actuated SMP artificial muscles.

## 1. Introduction

Natural muscles generate enough mechanical energy through enormous driving strain and rapid response to achieve complex movements such as running, swimming, climbing, and flying. Artificial muscles can match specific temporal, spatial, or force regimes typical of biological nature, but so far they cannot fully replicate all of these capabilities [1,2]. Thus, imitating natural muscles has been an essential challenge and opportunity for scientists. However, studies show that the performance of some artificial muscle materials exceeds that of natural muscle in some aspects. They are therefore particularly attractive for many applications where a muscle-like response is desirable, for-example in medical devices, prostheses, robotics, toys, biomimetic devices, and micro/nanoelectromechanical systems [3], in which, the high-loading actuators use electro- and thermo-activated artificial muscles [4,5]. Soft robots can also be used in an extreme biological environment with photo-activated motors [6,7]. Therefore, the field of artificial muscle is highly interdisciplinary and overlaps with various areas such as material science, chemical engineering, mechanical engineering, electrical engineering, and chemistry [3]. Compared to other materials, polymer materials have the advantages of easy preparation, low price, high elasticity, superiorities at large deformation and self-healing. Further, polymeric artificial muscle exhibits excellent multiple stimuli-responses [8,9,10,11]. Therefore, Mirvakili and co-workers [12] demonstrated a multidirectional artificial muscle from a highly oriented nylon filament that could bend and thus eliminate the need for a mechanical transmission mechanism and space for storing long linear actuators. The oriented nylon fiber with a thermal conductivity of 0.1 W·m^−1^·K^−1^ showed a 5% deformation with a temperature change from 25 to 140 °C. By cycling the input power, they observed that these nylon fiber actuators had a fully reversible amplitude response over 100,000 cycles.

Similarly, Chen and co-workers [13] developed aerial robots powered by multi-layered dielectric elastomer artificial muscles. The power density of the artificial muscle was 600 W·kg^−1^. Driven by the polymeric actuators, these robots performed a variety of flight motions, such as passively stable ascending and controllable hovering. Furthermore, they observed that these robots could detect and withstand collisions with obstacles. Arazoe and his co-workers [14] reported the development of a humidity-driven film actuator, consisting of a π-stacked carbon nitride polymer that was able to rapidly respond to the absorption and desorption of traces of water within 50 ms. Meanwhile, under the irradiation of 365 nm UV-light, the film quickly bent to an angle of 360° and jumped vertically up to 10 mm from the surface.

Although polymeric artificial muscles resemble to a large extent natural muscles, challenges remain within many of the technical applications. Dielectric elastomer actuators (DEAs) are capable of achieving periodic locomotion at high frequencies and generating high power densities; nevertheless, the weight of kilovolt supplies has limited the performance of DEAs [15]. The voltages employed in ionic artificial muscles are low, but they still suffer from high energies because of the close spacing between ions and electronic charges and the transfer of charges [3,16,17,18]. However, SMPs have clear beneficial properties, including their simple, fast response, high extensibility, and high power density with low voltage requirements. They are suitable as easily controllable actuators or heat-activated artificial muscles. Therefore, SMP-based soft actuators have been applied in many robot designs, such as soft swimming robots [19], sequential self-folding or interlocking components [20], hinges for the deployment of a solar array prototype in the aerospace industry [21], and rigid cable inflatable (RI) structures for large space systems [22,23]. Here, we focus primarily on the scientific aspects of shape-memory polymer artificial muscles, including their mechanisms, applications and challenges.

## 2. Shape-Memory Polymeric Artificial Muscles

Artificial muscle is a generic term used for a class of bio-inspired materials and devices that can reversibly expand, contract or rotate within one component due to an external stimulus (such as voltage, current, temperature, or light). These three actuation responses can be combined within a single component to produce other types of motion (e.g., bending, contraction on one side of the material, and expansion on the other side) [24,25]. Various techniques have been used to produce artificial muscles in the past. For example, rubber was used for pneumatic artificial muscles, in which a gas (for example, air) was used as the energetic source to control or expand the rubber bladder. For conductive polymer materials, electric-current or voltage was used as the deriving energy source [24,25,26]. However, for the actuation of shape-memory materials, an external stimulus (e.g., light, heat, or voltage) was used as a source of actuation. The artificial muscles based on these materials were termed as pneumatic/electro-active/shape-memory artificial muscles. All of these SMPs and artificial muscles were observed with specific limits. Deeper insight into the working mechanism of polymeric shape-memory polymers and artificial muscles is as follows:

### 2.1. Shape-Memory Effects

SMPs are stimuli-responsive smart materials that can undergo recoverable deformations upon application of an external stimulus. This phenomenon in SMP stems from a dual segment system, i.e., the cross-links that determine the permanent shape and the switching segments with a transition temperature (T_trans_) that fix the temporary shape. Below T_trans_, the SMPs remain stiff, while they become relatively soft upon heating above T_trans_. Consequently, they can be deformed into a desired temporary-shape upon applying an external force. While cooling and then subsequently removing this external force, their temporary-shape can be maintained for a long time. However, upon re-heating, their temporary deformed shape will automatically recover the original permanent shape. From here, it is clear that each SMP consists of dual-segments. One is highly elastic, and can be in molecular entanglement, the crystalline phase, chemical cross-linking, etc. The other is a reversible domain that determines the temporary shape and reduces its stiffness upon a particular stimulus. It is usually related to crystallization/melting transition, glass transition, reversible molecular cross-linking structure, etc. Upon triggering, the strain energy stored in the temporary shape is released, which results in shape recovery. The fixed and reversible domain determines the shape-memory performance of SMPs [8,22,27].

The SMPs can be divided into those showing one-way and two-way shape-memory effects (2W-SME), as shown in Figure 1. The one-way SME is irreversible, meaning that once the shape-memory process terminates, the SMP is fixed to a specific structure artificially to restart the shape-memory process. The performance of these SMPs is termed as the one-way shape-memory effect (1W-SME). In these SMPs, the transitions from temporary shape to permanent shape cannot be repeated by simply reversing the stimulus. Here the shape changing occurs only in one direction (Figure 1a). A new programming process is necessary every time to achieve the temporary shape (after the recovery process) [28,29]. These can be divided into dual-SMPs and multi-SMPs. If the SMP only remembers the temporary shape, then it is termed dual-SMP material. However, if the SMP remembers two or more temporary shapes, then it is termed a triple or multiple-SMP. In multiple-SMPs, the deformed material returns to its original shape step-by-step from two or more temporary shapes, as shown in Figure 1b.

Obvious realization of the multi-shape-memory effect (multi-SME) is mainly determined by two kinds of strategies. One strategy is to use polymers with a broad thermal transition in which multiple thermal transitions and temporary shapes are programmed at multiple temperatures across the broad transitions (with different composition materials). In this system, a predominant blend is prepared with broad glass transition that varies with the blend composition (due to miscibility). The other method to achieve broad thermal transition includes grafting, blocking copolymerization of different components, performing chemical cross-linking coupled with supramolecular bonding, etc. However, the important point to note is that it is very difficult to obtain a broad thermal transition with a chemical reaction (due to its complex nature). Further, the method based on miscible polymer blends is limited because most polymer blends are immiscible. This is why few research efforts have been dedicated to these types of SMPs [27,30]. The other kind of strategy to achieve multiple-SME is to construct several domains with well-separated thermal transitions. This method involves blending two chemically cross-linked polymers, copolymers or composites. In these blends, the reversible domains are related to the two crystallization/melting transitions, or one crystallization/melting temperature and one glass transition temperature of the polymers/composites [28]. This strategy to achieve multiple-SME is more exciting because SME is endowed by controlling the appropriate microstructure.

However, the two-way SME responds entirely to external stimuli, is reversible, and does not require additional programming of the material itself. Liquid crystal elastomers [31], cross-linked crystalline polymers [32], and their composites show these features [33]. The two-way reversible shape-memory effect can further be subdivided into quasi two-way and stress-free two-way shape-memory effects [34]. The quasi-2W-SME can be observed both in LCEs and semi-crystalline networks under an external stress. LCEs are elastic polymer networks containing main chain or side-chain liquid crystal units (LC-units). These LC-units are capable of undergoing reversible mesomorphic-to-isotropic phase transitions. In a liquid crystal these domains are typically randomly oriented with respect to each other, and thus are called liquid crystalline polydomains. During confirmation of the LCE network, these polydomains can be aligned when an external filed is applied (e.g., magnetic field or stretching force). This results in the alignment of the monodomain elongations of the LCE strip. When heated to a temperature above the liquid crystal clearing temperature (T_cl_), the polymer chains reduce its anisotropy. Hence macroscopic contraction of the sample occurs upon cooling below T_cl_, and the sample reverts to the original anisotropic state (elongation). This process is fully reversible and the monodomains can be formed either physically or by a chemical process using a two-step cross-linking process or a one-step cross-linking process [28,35,36]. During the two-step cross-linking process, in the first step, an isotropic network is established via partial cross-linking, then anisotropy is induced (via deformation) and fixed further by cross-linking in the second step [37,38,39,40]. However, in the one-step cross-linking process, small molecules or polymeric liquid crystal precursors are macroscopically oriented by applying an external field. After that, aligned precursors are polymerized/cross-linked to form a macroscopically anisotropic LCE [41,42,43,44]. However, in the physical process, the monodomain formation occurs via hanging an external weight or stress to the already synthesized LCE polydomains [45]. The basic difference between these two cross-linking methods is that the chemically cross-linked method cannot be altered after the fabrication process, while the physical fabricated network can be tuned easily (by applying external stress). This quasi-2W-SME can also be observed in semi-crystalline networks under a constant tensile load. The semi-crystalline network (of polycyclooctene) underwent elongation when it was cooled across the T_m_ (i.e., crystallization induced elongation or CIE). When heated above the T_m_ under the same load, the elongation reversed (i.e., melting induced contraction or MIC) [41]. The CIE-MIC transformation for the semi-crystalline networks requires the presence of an external force. Furthermore, the cross-link density is considered a tailoring parameter to control the quasi two-way shape-memory response [46,47,48,49]. If we look back at the mechanism of LCE, the anisotropic alignment of the polymer chain is the true inherent mechanism for the semi-crystalline polymer network. Although it is the external stress that is changing the anisotropy and strain change, the requirement of an external stress is a serious limitation for the application of potential quasi-2W-SMP devices. Therefore, the search for alternative mechanisms and materials to enable stress free 2W-SME has been a constant chase for the SMP community.

In this regard, Landlein’s group [33] synthesized a polyester urethane (PEU) network with a poly ω-pentadecalactone (PPDL) and PCL segment. The basic steps were similar in their mechanism to irreversible multi-SME (triple -SME) (Figure 1b), but no force was required for the cyclic actuation. The two polyesters provided a high T_m_ (T_m_,_high_) of around 64 °C and a low T_m_ (T_m,low_) of around 34 °C, respectively. The original shape of the PEU sample, i.e., shape S, was first deformed at a T_reset_ > T_m, high_ by applying an external force. This deformation was fixed by obtaining shape S_1_ at a lower temperature (T_low_), i.e., T_low_ < T_m_,_low_, while removing the external force, as shown in Figure 1c. At this point, the chain confirmation associated with PPDL changed. After this, the PEU sample was reheated again to the T_high_, i.e., T_m,low_ < T_high_ < T_m,high_, leading to another shape, S_2_. During this time the anisotropy and chain confirmation of the crystalline phase of PCL changed. Upon reheating to T_high_, the partial orientation in the PCL chain was removed and the deformation fixed by the PPDL domain remained untouched. This behavior sets the network anisotropy for the PCL domain without external force. Hence, macroscopic CIE-to-MIC transformation of PCL domains can be induced without external force. Overall, this internally created network anisotropy is reversible and differentiates the reversible 2W-SME from the irreversible multi-SME.

Later on, many attempts were carried out to achieve the stress-free two-way shape-memory effect to overcome the limitation of constant force by utilizing a variety of methods, such as bilayer polymeric laminate [50], core-shell composites [51,52], and crystalline polymeric multi-networks [53,54,55]. Along with these other methods, free-standing [56], autonomous [57], and controlled shape-memory actuation [58] methods were introduced by using a glassy thermoset-stretched liquid crystalline network, epoxy-based shape-memory lightly cross-linked network, and carbon nanotube/epoxy shape-memory LCE, respectively. The general design principle for all of the above free-standing two-way SMEs was the preparation of anisotropic networks (or built-in stress) that can provide an actuating force for reversible two-way SME without an external load.

Additive manufacturing is also gaining popularity in various scientific disciplines for device fabrication and tissue engineering [59]. The 3D printer can extrude a thermoplastic or LCE molten polymer that cools and solidifies to form a 3D structure, when cycled above and below their transition temperature or nematic-to-isotropic transition temperature (TN1). However, their development is limited to 3D-printable functional materials [60]. The thermoplastic/LCE ink with the highest triggerable dynamic bonds can lock controlled network configurations in the form of a 3D shape on exposure to UV light without an imposed mechanical field [59,61].

The 3D-printed reversible shape-changing soft actuators show 2W-shape-changing behavior. The printed conductive wires actuate LCE/SMP strips via Joule heating (UV light or heat treatment). The uniaxial deformation of the SMP/LCE strip acts as a driving force to achieve bending [62]. The 3D-printed shapes can be applied to flexible electronic devices, i.e., soft crawler, sensors, self-deploying devices, and implantable medical devices [62,63,64,65]. The 3D printing techniques are now evolving towards 4D printing, which has attracted increasing interest since its development. The materials used in 4D printing include hydrogels, multi-material shape-memory composites and LCEs. Unlike typical SMPs, 4D-printed materials can show a triple shape-memory effect. These triple shape-memory polymers possess two distinct temporary networks, which allows them to memorize an additional temporary shape [63,64,66]. The fundamental principle of 4D printing is to directly combine the structural design of the shape change to the material components and 3D printing processes. It can simplify the design strategy and fabrication process, and realize desirable 4D properties. Compared to the traditional manufacturing processes, such as molding and cutting, the 4D printing process can significantly save on fabrication costs. From all of this discussion, it is clear that the field of artificial muscles is strongly developing and more and more techniques are advancing the field of artificial muscles.

### 2.2. Programmable Shape-Memory Polymeric Artificial Muscles

Artificial muscle applications require a stimuli to generate arbitrary three-dimensional SMP shape changes. These shape changes can be achieved by utilizing “programming processes” [67]. The programming process is the manipulation of external, physical processes that determines the shape-changing pathway. This process is independent of material fabrication and is based on the molded system, so that it can occur precisely according to the demand. The shape shifting performance of any SMP is accompanied by a force generation, which converts the stimuli into mechanical energy. With the programming process, the sample can perform remote self-locomotion as a robotic. By cyclic stimuli, the reversible bending and unbending can be transformed into walking, swimming and load lifting abilities. As shown in Figure 2a, the Wang group [40] made significant progress in the preparation of PCL- and polydopamine (PDA)-based shape-memory polymers. They utilized two kinds of lactones as raw materials with similar but different thermal properties. They incorporated very small amounts of polydopamine (PDA) nanospheres into a poly (ε-caprolactone) semi-crystalline copolymer network and PCL-co-pentadecalactone (PCL-co-PDL) prepolymers. The addition of PDA nanophores showed a profound effect on the melting and crystalline enthalpies of the PCL-co-PDL segments, which were three times as much as those of the polymer composites, indicating the complete inhibition of PDA nanophores inside the polymer network without affecting the crystallization of the homopolymer (PCL). The PDL segment served as a geometric frame contributing to the main melting point. The melting temperature was used as an actuation temperature. It was selected (T_m_s) in between the melting temperature of two polymers within a range of 50–65 °C. This is because the melting temperature of the PCL segment was in the range of 15–50 °C, whereas that of the PCL-co-PDL segment remained in between 65–80 °C. The whole procedure was divided into three steps. To begin with, when the temperature was higher than T_m_s, i.e., T_high_ > T_m_s, the molecular chains were in a viscous state and the material was easy to deform. After that, lowering the temperature to T_low_, i.e., T_low_ < T_m_s, the material returned to its original shape. Further, to observe the 2W-SME, the sample was first programed to a “V” shape (with an angle of 70°) in a water bath at 90 °C. The short arm of the programmed sample was fixed on a clamp. When the temperature was turned on (light on) the sample started opening and reached an angle of 115 °C (in 8-to-9s). However, when the light was turned off, the sample started closing to a smaller angle of nearly 90° and 70° in a time of 35 s and 83 s, respectively. Consequently, the temperature between T_high_ and T_low_ was used to make the process more programmable and conversable, due to the partial crystallinity (in the polymer). Apart from polycaprolactone, several shape-memory hydrogel materials have also been observed as programmable artificial muscles, and a typical example is displayed in Figure 2b. That is, a bilayer hydrogel was obtained with an asymmetric upper critical solution temperature (UCST). When the temperature was lower than UCST, the layer of poly acrylic acid and poly acrylamide were able to shrink, while driving the sample to bend on account of the hydrogen bonding. As the ratio of the two layers changed, the size of the changes could be programmed. Further, it was observed that photomask technology can be used to design interpenetrating network domains to achieve complex two-dimensional and three-dimensional deformations of the hydrogels [68]. Liquid crystal elastomers can also be used to create artificial muscles as shown in Figure 2c. Here, specifically, a new dynamic network based on reversible siloxane exchange reactions is shown. The siloxane liquid crystal elastomer was swollen in a solution of an anionic catalyst (TMA-DMSiO), and the siloxane exchange was induced at a specific temperature (100 °C) to establish a more complex motion mode or three-dimensional shape-change. Because the catalyst would disable at a high temperature (150 °C), a heating method was designed to stop the exchange reaction. For further research, by combining the photo-thermal conversion, different types of motion modules were integrated, while achieving a continuous NIR-induced process of grasping and transporting objects by software devices [69].

This means that the programming process, crystallinity and anisotropic behavior of the materials are the key factors that empower SMP materials to exhibit an artificial muscle performance. Along with this stimulus is another significant factor that is acting as a trigger to actuate the SME. Here we will focus primarily on the understanding of scientific aspects of programmable (both 1W and 2W SMPs) shape-memory artificial muscles, developed in the last 5–6 years. We do not focus on the artificially made biological muscles for in-vivo use in the human body. Therefore, the terms actuators and artificial muscles are used interchangeably throughout the manuscript.

## 3. Applications of Shape-Memory Polymeric Artificial Muscles

### 3.1. Thermo-Induced Shape-Memory Polymeric Artificial Muscles

The development of thermally actuated SMPs has focused primarily on relatively low temperatures (T_c_ < 100 °C) and using elastomeric polymers such as thermoplastic polyurethane (TPU), cross-linked polyethylene, polycaprolactone (PCL) and polynorbornene. These materials were considered appropriate for biomedical applications, such as smart fibers, shrinkable tubes, and aerospace applications using changes in modulus and switching temperature for shape change as well as actuation. These materials consist of network points and molecular switches that are constructed either by physical cross-links of intermolecular interactions or chemical cross-links through covalent bonds. This cross-linking structure leads to phase separation and produces hard segments, soft segments, and domain formation in the polymeric network. Inside this system, the hard segment serves as a pivoting point for shape recovery movement, while the soft segment mainly serves to absorb the external stress that is applied to the polymer [8,70]. Based on intermolecular interactions, these polymers can be further sub-divided into linear and branched polymers. For linear polymers, the shape-memory effect is due to their phase separation and domain orientation in the block copolymers such as PUs and PMMA-g-PEG copolymers. In polyester–urethanes, the oligomer segments serve as hard-segments, while the polyester serves as a switching segment (Figure 3). However, as for PMMA-graft-PEG copolymers, the PMMA is not covalently cross-linked with PEG, but rather the junction point of the backbone and side chains behave as a physical cross-linking point. On other hand, entangled branched copolymers usually take much longer to disappear by the repetitive motion of polymers [22,71]. Further, these materials have the ability of rapid shape recovery and demonstrate a durable elastic nature with bio-compatibility, which is used in biomaterials and shape-memory polymer textiles.

For preparation, the covalently cross-linked network points can be obtained by cross-linking of linear or branched polymers, as well as by copolymerization of one or several monomers, whereby at least one is considered to have at least one tri-functional group. Sometimes radical initiation reactions can lead to cross-linking in co-polymer networks [72]. Other synthetic routes rely on copolymerization of polymer networks by Diels–Alders reactions, or a two-step poly-condensation process, which not only improves the mechanical and shape-memory performance of the polymer composite, but also increases the durability during repeated actuation by promoting self-healing [73,74,75]. Here we provide a list of information related to the synthesis and internal mechanisms of some SMP artificial muscles so that readers can refer to relevant literature that discusses the synthesis of SMPs such as polycaprolactone-epoxy-based polymer networks, bisphenol-A epoxy resin [76,77], polyvinyl butyral-based polymer networks [78], PCL and poly-L-lactic acid [79], polycaprolactone-based polyurethane [80], PU/montmorillonite-PMMA composites [81], cross-linked poly(ethylene vinyl acetate) and poly(ε-caprolactone) [27], polyethylene glycol-based polyurethane (PEG-based-PUs) [82], polycaprolactone-based SMPs [83], and poly(ethylene oxide-co-ethylene terephthalate) [84].

Application environment, performance stability and cost problems have restricted the deployment of artificial muscles in robots, exoskeletons, miniature actuators for microfluidic laboratories, and prosthetic limbs. These materials are expensive in the sense that they need a sizeable driving force, which is challenging to control. Recently SMP-based thermal actuated artificial muscles, known as twisted and coiled polymer actuators (TCA), were prepared from nylon 6,6 fibers [85]. To some extent, these have promising potential because of their low cost, high strength, and reversible thermal expansion with large dimensional anisotropy. The composites are designed as a particular twist-insertion device that can be operational at a temperature of 30–150 °C with a load of 1400 g. Based on the TCA concept, Cho. et al. [86] synthesized an artificial finger that could successfully lift different objects as shown in Figure 4a–c. Similarly, Wu et al. [87] prepared a polyamide muscle-based artificial hand. It was comprised of a silicon tube (8 mm in diameter and 112 mm in length) with a pre-strain coiled fishing line and a spring through which hot and cool water was allowed to move freely during relaxation and contraction. The coiled fishing line muscles acted as a contractile element of the actuating system with 3.33 mm outer coil diameter and 86 mm length. On one end, a string was attached to the actuating system with a fingertip on the far end and the returning spring was used as a contractile element in the actuating system, as shown in Figure 4d. The flow of water was analyzed and set with existing computational software so that the system remained safe and workable. Figure 4d illustrates how, when hot water passed through the muscle, the nylon fiber showed a negative axial thermal expansion and sizeable positive radial thermal expansion, which resulted in contraction of the coiled fiber and hence the filament moved and lifted the weight (which was 200 g for 37.5 mm displacement).

In contrast, the flow of cold water through the spring resulted in contraction of the system that brought the system back to its original position. Hot water was supplied to the silicon inlet at 1.5 s and stopped after 7 s, followed by cold water until the actuating system returned to its original state. The cooling time was less than 5 s, and depended on the convective heat transfer, polymer heat capacity, and volume and space area of the muscles, since pure thermally actuated SMPs have low work capacity. Hains et al. [85] created novel shape-memory artificial muscles reinforced with CNT-fiber. The prepared muscles matched the performance of mammalian skeletal muscles and had nearly 20% tensile lifting ability under rapid loading. Precursor fibers were used to generate highly oriented polymer chains of polyamide and polyethylene (in the fiber direction), which had small negative thermal expansion co-efficients that yielded significant reversible contraction when the heat was applied. The already-induced twist brought chirality into the CNT fibers and inside the system. Polyamide 6, 6 fibers showed a 34% increase in reversible thermal contraction (from 20 °C to 240 °C) while the polyurethane monofilaments showed an increment of 16% for coiled muscles between 20 °C and 130 °C. The intercooler contact helped to build stiffness into the coiled structure along with increasing temperature, and produced a 24-fold increase in the related tensile modulus. Further, the tensile strain and load-bearing ability could be varied by adjusting the coil spring index (the ratio of mean coil diameter to fiber diameter), which was inversely related to spring stiffness. The maximum optimal load observed for the largest coil diameter with a spring index (C) of 1.7 was 22 MPa, with a 21% maximum strain. In contrast, with the smallest coil diameter with a spring index of 1.1, the optimal load increased to 50 MPa, while the maximum stroke registered was 9.3%. However, the maximum specified work during contraction for the coil spring of polyamide 6, 6 muscles with a spring index of 1.1 was 2.8 KJ/Kg, which was 64 times that of natural muscles. The average mechanical output power during contraction was 27.1 kj/kg, which was 84 times the peak output of mammalian skeletal muscles.

Similarly, we prepared shape-memory PUPCL copolymer materials [4] with self-healing abilities, which showed a high cycle life. From load lifting experiments, it was observed that these materials were able to lift a load of more than 20 times the mass of the actuator material (Figure 5a). More recently, another type of shape-memory polymer was prepared [88], which was synthesized from PCL, PDMS, and PUs. Along with the high cycle life (based on self-healing behavior), it could also lift a mass 500 times its weight within 5 s and the maximum power density registered was half of the mammalian skeletal muscles (Figure 5b). Moreover, Xie et al. [89] introduced poly (ethylene-co-vinyl acetate)/graphene (cEVA/G) shape-memory actuators. They developed a series of EVA-carbon fiber based composites (EVA/CF) with a remarkably enhanced recovery stress both in a free state and under compressive stress. The addition of CF into EVA increased the modulus and the recovery stresses. Consequently, cEVA/CF composites exhibited a robust shape recovery performance under a counteracting load. This behavior of the composite was well modelled in a deployable device, as shown in Figure 5c. Recently they observed that these materials had a cyclic and dual sensitive (light/thermal) capacity [90]. The cyclic actuation was enabled by crystallization-induced elongation and melting-induced contraction, which was induced by the EVA part of the composites. When using NIR (near infra-red) irradiation and direct heating, an indicating circuit, which used a lamp as an alarm, was enabled to form a conceptual actuator for sensing applications. This actuator could effectively raise a signal (i.e., the lamp is turned on) when responding to a direct heating source, and after removing direct heating, the alarm was disengaged, i.e., light is turned off, and elongation/shape fixation occurred at room temperature. The process repeated itself during shape recovery when the system was heated for recovery to complete the process.

### 3.2. Photo-Induced Shape-Memory Polymeric Artificial Muscles

Light-responsive molecules can undergo isomerization in the presence of a particular light stimulus that reversibly changes their structures between two or more chemical aggregates. To be more specific, this kind of isomerization affects the orientation and arrangement within the molecular chains. These changes express themselves in terms of color change or visible deformation on the macro-scale. In recent years, many scientists have prepared a large number of artificial muscles with a specific mechanical strength and light-driven ability by adding organic phases to the polymeric systems, e.g., azobenzene [91] or spiropyran [92], or inorganic phases, e.g., graphene [93], or metal nanoparticles [94,95]. In comparison to thermal-responsive polymers, light-induced polymers have the unique advantages of instantaneous control, environmental friendliness, non-contact initiation, etc., which is of importance in the aerospace, biomedical, and other fields.

In the past years, the most commonly used representative material that responds to light is undoubtedly the azobenzene group [96]. The polymers based on amorphous azobenzene have a 1% response (shrinkage) under ultraviolet irradiation [97]. Researchers have found out that aligned polymers, such as liquid crystal polymers, can improve the driving performance of photo-isomers. In addition, the combination of the entropy, the elasticity of polymers and the photo-isomerism of azobenzene is beneficial for obtaining potential artificial muscles with a large range of reversible deformation [98,99].

The reversible phase transition uses the trans–cis photo-isomerization of azobenzene groups (Figure 6a). The rod-like trans-structure of the liquid crystal phase is stable at room temperature, and the curved cis-structure is easily obtained under external stimuli. Typically, UV irradiation is used to trigger the isomerization phenomenon. For example, the isomerization of a single molecule with shrinkage in its length of 3.5 Å usually starts with ultraviolet irradiation. This isomerization is reversibly restored with visible light radiation as well as heating. Therefore, to obtain a unique controllable deformation, an azobenzene group was added to the polymer network [100]. The continuous accumulation of this isomerism and exposure to ultraviolet irradiation leads to a polymer phase transition. Thus, the final product changes from the anisotropic state to the isotropic state, while causing macro-scale deformation (Figure 6b). On the other hand, the photo-thermal effect can also induce isomerization in the system. This can be achieved by adding appropriate photo-thermal conversion particles, such as carbon materials [101], metal nanoparticles [102], and organic dyes [103]. Therefore, on the exposure of the compound to visible or NIR irradiation, the doped particles serve as a heat source to raise the temperature of the system above the liquid crystal transition point, resulting in the transformation to the isotropic state (Figure 6c). Furthermore, because of the Weigert effect, when the transition distance (long axis) of the trans azobenzene molecule is parallel to the polarization direction of the linearly polarized light, the azobenzene unit will absorb energy to reach the excited state and undergo trans–cis isomerization (Figure 6d). In contrast, the molecules that are perpendicular to the polarization direction do not undergo this process and maintain their initial state. After repeating multiple cycles of trans–cis–trans isomerization, the transition distance of all trans-azobenzenes will be perpendicular to the direction of polarized light. Subsequently, it will become stable and the photo-reorientation of azobenzene will be complete (Figure 6d) [102].

The spray-coating technology was used by Schenning’s group [104] to combine the polyethylene terephthalate (PET) and liquid crystal network. Firstly, the thermoplastic PET was shaped into arbitrary shapes (such as origami-like folds and spirals) by heating. Then the heat-fixed geometry formed different shapes reversibly by irradiation with ultraviolet light. The whole process yielded a final structure that had a substantial and reversible driving force and took complex forms through winding, curling, and unfolding. With flexible shape reprogramming ability, the same sample was redesigned multiple times in favor of manufacturing a mechanically robust, recyclable, and light-responsive actuator with a highly adjustable geometry (Figure 7a) [105]. Zhang and his co-workers [106] made use of the mismatch of the coefficient of thermal expansion between graphene oxide and an azobenzene-doped liquid crystal network to design a delicate bilayer soft actuator through the micro-channel method. In Figure 7b, the bilayer can be observed. It reacts with both UV and NIR exposure simultaneously due to the addition of azobenzene and graphene oxide, respectively. This structure is expected to be used widely in bionic and intelligent soft robots [107]. Similarly, Zhao et al. [108] used a reprogrammable azobenzene-containing liquid crystal network to demonstrate a new strategy for improving the light-induced mechanical actuators. These actuators were designed in the form of wheels and spring-like “motors” with adjustable rolling or moving directions and speeds (Figure 7c) [108]. Furthermore, by incorporating a type of fast trans–cis azobenzene derivative into a liquid crystal network, Gelebart and Mulder [91] produced a photo-sensitive polymer film with aligned and macroscopic mechanical waving behavior (Figure 7d). Their group also presented theoretical models. They used several photo simulation methods and explained the mechanism of the generation of the wave. The fundamental idea was to store the mechanical strain energy in the polymer in advance. After that, the UV light was used to convert luminous energy to mechanical energy via photoisomerization by triggering the preloaded strain energy. Furthermore, potential applications of photo-induced films in light-driven motion and self-cleaning surfaces were proposed, involving photo-mechanical energy and miniaturized transportation.

There is also a range of other functional groups that possess photoresponsive behavior. The spiropyran molecule is distinctly sensitive to ultraviolet light and has been used to make a molecular switch (Figure 8a) [109]. The spiropyran structure shows three different light-response properties at three different wavelengths. The cis–trans isomerization was observed under 405 nm irradiation. The decomposition of the 2-nitrobenzyl group was observed with 254 nm irradiation, and a spiro-mesocyanine isomerization occurred under 365 nm light. This makes the spiropyran a promising candidate for the further design of light-responsive polymeric materials [109]. Similarly, Zhao and his co-workers [110] reported a dye-doped liquid crystal polymer containing a metal-bis-dithiolene complex as a molecular photothermal agent. In their article, they observed that with two ends of a strip fixed, a NIR laser could generate a wave that pushed a rod forwards or backwards. However, when only one end was fixed, the soft actuator performed various autonomous arm-like movements depending on the direction and angle of the laser beam (Figure 8b). Brannum and co-worker [111] designed a well-aligned backbone liquid crystal elastomer with a cholesteric phase. Due to the introduction of heterogeneous photosensitive chiral olefin in the molecule, the selective reflection bandwidth of the polymer increased to more than 200 nm. At 200 °C, they observed a deformation and discoloration (Figure 8c). Jin and Song [112] presented a new strategy that involves a combination of a programmable crystalline polymer and reversible optical bonds to achieve the function of artificial muscles. The integral three-dimensional structure was based on the plasticity-based origami technique, with nitro-cinnamate, with photo-reversible dimerization, added into the system to program the geometric changes as shown in Figure 8d. This method represents a general approach for creating photo-induced shape-memory polymeric artificial muscles [112]. It should be noted that there are a substantial number of other molecular systems that have not been mentioned here, which have the potential to be integrated into the development of photo-induced shape-memory polymeric artificial muscles.

Ma et al. [113] created a new design based on a mixed-matrix membrane strategy to obtain photo-induced SMP artificial muscles that lift weights. The formed hybrid system bridges the gap between the fast light response and the suitable elastomer properties with a high Young’s modulus. The designed artificial muscles perform a variety of functions, from the ability to hold something similar to human hands, to lifting weights (Figure 9a). What is more interesting is that Dicker’s group [114] proposed a new strategy to solve the insufficient driving performance at the molecular level (Figure 9b). They successfully achieved a three-fold chemical amplification of the driving force by using the combination of actuation and a light-sensitive acid autocatalytic solution that activates and deactivates at a specific light wavelength.

### 3.3. Electro-Induced Shape-Memory Polymeric Artificial Muscles

Heating above transition temperatures, such as T_g_ or T_m_, is usually used in shape-memory materials. Therefore, to obtain uniform heating, electrically conducting composites were synthesized by using CNTs, graphene oxide, a CNT membrane, and carbon black as fillers in which the applied voltage yields an electrical pulse and initiates actuation and deformation of the material via Joule heating [115,116,117,118,119]. Thus, the primary source of mediation is the applied field, which provides potential and brings changes in the soft segment of the polymer to restore its original state. This means that the underlying concept required for the shape-memory phenomenon is the same in electro-active (electrically actuated) shape-memory polymers as in thermo-active shape-memory polymers [120,121]. Consequently, these materials have received great interest because of their successful applications in electro-active activators, such as in smart actuators and micro aerial vehicles [122]. The mechanical reinforcement and functionalization of these filler-based SMPs are achieved mainly through the hybridization process. Generally, chemical hybridization is considered superior to physical blending because it can improve interfaces between the polymer and fillers via a fine dispersion process. Furthermore, chemically incorporated fillers provide multifunctional cross-links, which not only augment rubber elasticity but also enhance conventional strength and strain recovery [119,123]. The reason for this is that the CNTs have unique structural arrangements of atoms, a high aspect ratio, and excellent mechanical, thermal, and electronic properties. Additionally, CNTs are highly flexible, which gives them remarkable advantages, making them the best reinforcement component in host polymer matrices [119,124].

Similarly, graphene, which is essentially a CNT cut along its axis so that it unrolls and lays flat, has 2D sp^2^ hybridization carbons and exhibits excellent electrical conductivity. Further, the 3D interconnection of graphene can be obtained by freeze casting, self-gelation, and chemical vapor deposition. In polymer foams, the graphene not only increases conductivity but also contributes to the improved mechanical properties [123,125,126,127]. Furthermore, carbon fiber and oxidized graphite also show high electrical conductivity because of their high surface area and surface polarity. Using this electrically induced actuation mechanism, Liu et al. [128] studied an electro-active shape-memory composite of a CNT/graphene aerogel (Figure 10a). When CNT and graphene, in a weight ratio of 3:5, were added to an epoxy resin, the electrical conductivity was nearly 16.3 S/m and the composites showed shape recovery after 120 s at a potential difference of 60 V. Mohan and co-workers [129] combined poly (lactic acid) with CNT and obtained a nanocomposite by a normal chemical process as shown in Figure 10d. The nanocomposite had a conductivity of 10^−6^ S/m with shape recovery behavior within 11 s at a constant voltage of 60 V. The same phenomenon was observed for a composite of Cu-decorated CNTs dispersed in PLA/ESO (epoxydized soybean oil). The nanocomposite recovered to its original shape within 35 s with a voltage of 40 V (Figure 10b) [130].

A remarkable shape recovery behavior (within 12 s) was observed for a poly (ethylene-co-vinyl acetate) /Poly(ε-caprolactone) /CNT (EVA/PCL/CNT) blend (Figure 10c). Recently, various synthesis methods were studied for the preparation of electro-active shape-memory polymers, based on the following fillers: graphene oxide [131], carbon black [132], CNT layers [133], TiO_2_ [134], carbon fibers [135], and single-walled CNTs [136]. Most of these fillers not only made the polymer material conductive but also improved the mechanical properties. Most of the materials showed a shape recovery performance within 60–100 s with an applied voltage of 16–70 V. However, none of these materials showed load-bearing properties. Nevertheless, Yip. et al. [137] prepared high-performance robotic muscles from conductive polyamide 6,6 sewing thread. The nylon 6,6 sewing thread was made conductive by coating it with electric heating, while using the same method as Haines et al. [85]. By using this concept, they prepared a robotic hand with 3D-printed ABS (Acrylonitrile Butadiene Styrene) material. A flexural design with conduits for a tendon was used with supercoiled polymer (SCP) actuators on each tendon (for actuation).

These actuators provided a strain of approximately 10–15% to produce a full range of motion. To mimic the physical location of these muscles in a human arm, they spread them along the forearm of the robot. Further, to cool the actuators during relaxation, four computer fans were used. The power to weight ratio demonstrated by these muscles was 5.38 kW/kg, which is 17 times the power to weight ratio of mammalian skeletal muscles. The grasps were performed within a second without the need for a feedback sensor and with no noticeable crosstalk between actuators. Most recently, Peng’s group [138] prepared a 3D porous network composite material (Figure 11). The porous nanotube served as a built-in integral conductive network, which provided homogenous in-situ Joule heating for the composite polymer. By using this basic concept, they used the material to prepare an inchworm-like robot. The basic design of the robot was that thin metal plates were fixed to both sides of the polymer composites as legs to increase the size of each step (Figure 11). They designed a particular track with a metal sawtooth-like structure. The locomotion of the robot involved releasing and grasping of the front and rear sets of legs. This is because when the robot releases its legs, the rear legs are then stuck to the metal sawtooth, and the front legs push the robot in the forward direction. After grasping, the front leg is hooked on the sawtooth and pulls the rear leg forward. This movement is repeated continuously, in analogy to inchworm locomotion, by cyclic releasing and grasping of the front and back legs. The total time required for a complete cycle was 120 s with an alternating voltage of 2 V and 8 V. The locomotion observed for the inchworm robot was 8 mm in a total time of 10 min, which further increased to 1.2 cm in 10 min when the cycling time was set to 40 s (Figure 11c). These robots were considered to have profound potential with numerous advantages, such as their simple structure, light weight, and low cost, with designable parameters including speed, frequency, and length.

Wang et al. [7] repeated the same process while preparing a self-adaptable, entirely soft-bodied electronic robot from LCE-CB nano-composites. The robot was able to achieve effective soft-bodied locomotion based on programmable body bending and anchored motion similar to that of an inchworm. Their robot was superior in regards to movement, sensing in both the forward and backward direction, and response times. The demonstrated soft robots were capable of sophisticated shape adaptation, and two-way gait locomotion in programmable and adaptive sensing actuation manners. Further, this process involved a soft Joule heating electronic mesh, ultra-thin Si optoelectronic sensors, and thermal responsive LCEs. Kim et al. [139] developed a strategy by arranging CNTs in the LCE network. The arrangement of these CNTs not only enhanced the mechanical performance or electrical conductivity of the LCE network but also served as an alignment layer for LCEs. By controlling the location orientation and quantity of layers of CNTs in LCE-CNT composites, programmed and patterned actuators were built that respond to electrical current. The response of these actuators to a DC voltage of nearly 15.1 V/cm registered a 12% actuation strain and a work capacity of 100 kJ/m^3^.

Similarly, Sun et al. [140] prepared electro-active actuators (EAAs) by utilizing super aligned carbon-nanotube sheets with poly(dimethyl siloxane) (PDMS) layers. The programmed EAA was capable of deforming to a bending angle of 540° at 12 V. This concept was further utilized by fabricating a small crawling robot that mimicked worm-like behavior under a cycling voltage (due to Joule heating). The EAA started to expand from 0 to 20 s once the voltage was activated. After the voltage was switched off, the EAA returned to its curved state due to fast heat dissipation. This allowed the EAA to move forward on the rigid rail.

However, Xiao et al. [141] prepared soft robots (Janus flower-like structure) by using a liquid crystal polymer (LCP). They used the basic concept of the order–disorder phase transition of mesogens in the oriented liquid crystal network. They designed soft robots from uniaxially oriented LCN (liquid crystal network) strips, a laminated Kapton layer and embedded thin resistive wires (in between), as shown in Figure 12a. The LCN layer served as the active layer. In contrast, the thermo-stable Kapton layer film acted as the passive layer. The concept of Joule heating was used to induce contraction of the LCN, which triggered the deformation of the actuator. They used programming conditions, as shown in Figure 12b, where the passive Kapton layer displayed a certain plastic deformation that helped the LCN with oriented mesogens to retain the programmed shape at room temperature (shape A). However, under electric power (voltage on), the contraction force of the LCN with mesogens in an isotropic state overcame the elastic force (of the deformed Kapton). It thus displayed a shape change (shape B). When the electric power was off, the extension force of the LCN with re-oriented mesogens worked with the recall force of Kapton and brought the ELCN(LCN-heating wire-Kapton actuator) back to a programmed shape (Figure 12c). By using this basic concept, they designed the ELCN actuator, as shown in Figure 12d. The design of the ELCN gripper was similar to an elephant trunk. The ELCN gripper gripped (when the power was on) and released (when the power was off) objects of different shapes and weights, such as a syringe (4.2 g), a tube of water (5.3 g) and a string of electric coils (7.4 g). As the weight of the prepared ELCN was only 35 mg, the load lifted by the actuator was 210 times heavier than the actuator.

Similarly, He et al. [142] prepared an electro-active LCE-based soft tubular actuator. The benefits of the tubular actuator include multiple actuation modes, contraction, bending, and expansion. The LCE-based tubular actuator was prepared by sandwiching three separate thin stretchable serpentine heating wires between two layers of loosely cross-linked LCE films (Figure 13a). After sandwiching, the whole structure was exposed to UV irradiation to fix the alignment of the liquid crystal mesogen. The heating wires were used to control the LCE actuator, i.e., to bend or contract the actuators by Joule heating (Figure 13c). After performing various experiments, i.e., exposing the LCE actuator to an applied potential that derived actuation of the LCE thin film, they used it to prepare soft grippers, as shown in Figure 13b. The three tubular actuators were first attached to a circular plate, which was further connected to the micro-controller to control the actuation electrical potential. By selectively actuating heating wires in each tubular actuator of the gripper, it was able to grasp and lift a vial after twisting its cap without additional external control. Recently, Xu et al. [124] prepared a mechanical gripper made of poly(ethylene-co-octene) (PEO) and segregated conductive networks of carbon nanotubes (S-CNT) (Figure 13d). The PEO-CNT composites with segregated structures had a low modulus and high conductivity, as well as a fast response at low voltage. The actuation mechanism was realized by the shape-memory behavior of the PEO, which is based on the crystallization of ethylene sequences in PEO and the crystallization and melting of PEO-CNT composites. Based on this concept, they prepared a gripper that was able to grasp the objects through an electro-active process. As shown in Figure 13d, the grippers were able to open their fingers within 18 s with a voltage of 36 V and close them within 168 s with the voltage off, with 186 s needed for the entire cycle.

Similarly, Rubaiai et al. [143] prepared a self-pneumatic actuator (SPA) by using the shape fixation and shape recovery behavior of conductive poly-lactic acid (CPLA). The fabrication of the SPA was such that the CPLA was integrated into the flat side of the SPA. The components of the SPA were fabricated using casting and 3D printing. However, local indentations, i.e., joint 1, 2, and 3, were designed in the flat sheet geometry to facilitate bending at the hinge locations. For Joule heating, silver wires were soldered to each hinge without affecting the device flexibility. After this, the CPLA/silver wire was encapsulated through a silicone rubber bath to allow adhesion with the SPA. When Joule heating is activated at a given location on the CPLA, the material nearby softens and thus enables bending at that location. Using this concept, they prepared actuators that enabled them to grasp multiple objects using different grasping modes. The maximum load held by the SPA was 800 g when the SPA was actuated with 22 psi and all CPLA joints were actuated with 12 V inputs.

## 4. Concluding Remarks with Future Perspectives

Exciting properties of SMPs have endowed these materials with desirable utility for artificial muscle applications. In the past, most of the applications were based on SMP actuators. However, along with the development of SMP programming methods, cross-linking methods (in various combinations) and different trigger methods suggest that more applications in the form of artificial muscles, robotic fingers, deployable devices, and robotics can be realized. Fundamentally, SMP artificial muscles are providing a new alternative to natural muscles by imitation, which inspires and challenges the material scientists to untangle the structure–property and application relationships. It is intriguing to see that the artificial muscle performance in some cases exceeds that of natural muscles, which depends not only on the intrinsic properties of these materials but also on the actuation mechanisms, miniaturization and specific design. Aside from the significant progress in SMP artificial muscles, some continuous effort should be put into the following aspects of SMPs and composites, which have been mostly unexplored to date.

Among existing SMPs, polycaprolactone, polyurethane, specific epoxy resin and LCEs are often used as artificial muscles, presumably because these materials have a crystalline domain, which induces the shape-memory phenomenon. Exploration of new polymer systems with similar properties to these materials may lead to new functionalities and applications in the field of artificial muscles.

It is generally accepted that SMP artificial muscles usually mimic human muscles, but there is a lack of SMPs that mimic the strength associated with the actuation of humanoid muscles. Therefore, appropriate methods are needed to quantitatively assess the contribution of different factors that affect the mechanical performance of polymer-based actuators.

A large effort has been devoted to controlling the load-bearing performance of SMP artificial muscles, but little is known about how to develop millions of reversible contractions with rapid load-bearing abilities. Distinguishing different SMP networks would help greatly in understanding better the structure–property relationship and reversible relaxation abilities with maximum cycles.

Current knowledge of SMP artificial muscles is generally based on shape-memory polymer networks and their composites with thermal or electrical actuation. However, little is known about LCEs and their regular systems. Therefore, thorough study is needed to address many questions related to LEC-based SMP artificial muscles, activated either thermally, electrically, or by light.

Limited fatigue resistance and cracks on the surface of these materials is a significant limitation at the current stage. Therefore, the combination of these materials with self-healing materials may provide a possible solution to the development of new self-healing or self-recovery SMP artificial muscles.

Combined with the load lifting performance, the development of new multifunctional artificial muscles with opto-active or magneto-active shape-memory performance in a miniaturized form is still a big challenge. Despite many attempts for the most common applications of SMP artificial muscles, the actual arrangement of these in robotic design is one of the highly essential issues that need to be taken into account.

Furthermore, the electrical actuation of the electro-active SMPs is carried out by dispersion of fillers, for example, CNTs in the polymer network. Sometimes, weak linkages of these fillers in a polymer chain may induce defects that significantly reduce the strength. Therefore, better annealing and optimized synthesis conditions are required to overcome these defects and junctions between the nanotube yarns in the polymer networks.

How to effectively dissipate the heat when the SMPs are restored to the original shape is also a big challenge. This is critical to load-bearing two-way SMPs with many shape-changing cycles. Hence, the investigation of SMPs should be combined with new technologies of thermal management, which will bring SMP research closer to real applications.

## Figures and Tables

**Figure 1 molecules-25-04246-f001:**
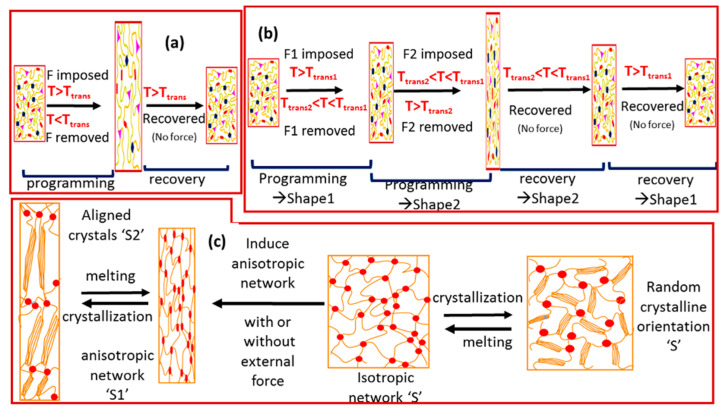
Schematic illustration of (**a**) dual shape-memory, (**b**) multiple shape-memory effect and (**c**) mechanism for 2W-SME of the cross-linked crystalline polymer network with an isotropic and anisotropic network structure (reproduced with permission [28]. Copyrights (2015), Elsevier).

**Figure 2 molecules-25-04246-f002:**
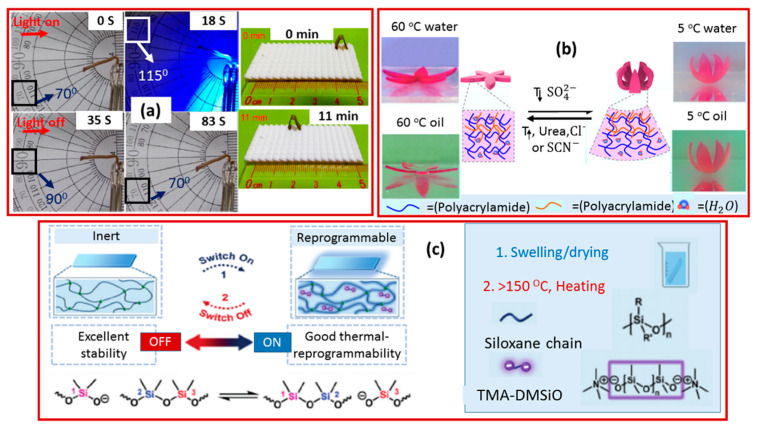
Various programmable shape-memory polymeric artificial muscles based on (**a**) PCL and polydopamine (Reprinted (adapted) with permission [40]. Copyright (2018) American Chemical Society); (**b**) hydrogel (Reprinted (adapted) with permission [68]. Copyright (2019) American Chemical Society); (**c**) liquid crystal elastomer (Reprinted (adapted) with permission [69]. Copyright 2020 John Wiley and Sons).

**Figure 3 molecules-25-04246-f003:**
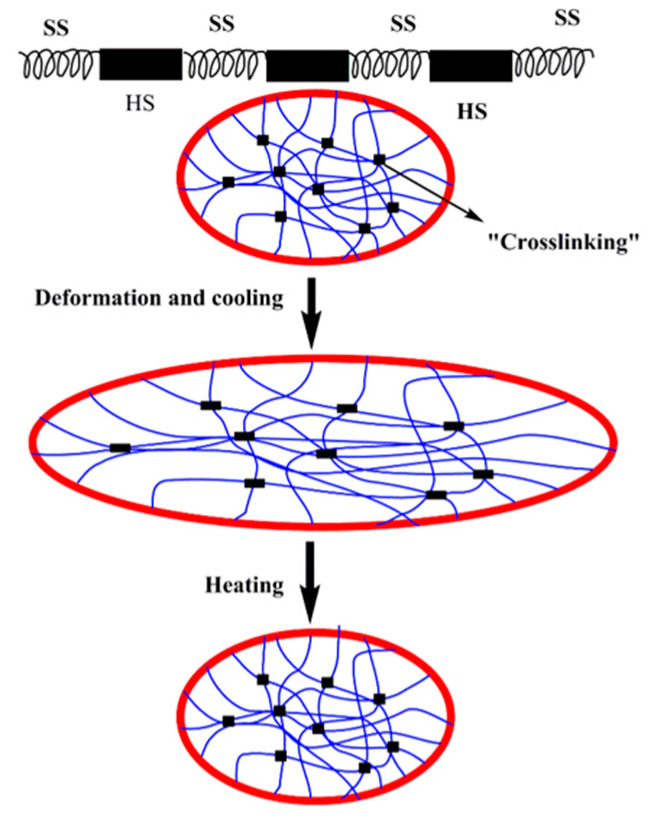
The micro-mechanism of the shape-memory effect of polymers, where HS refers to the hard segment, while SS refers to soft segments.

**Figure 4 molecules-25-04246-f004:**
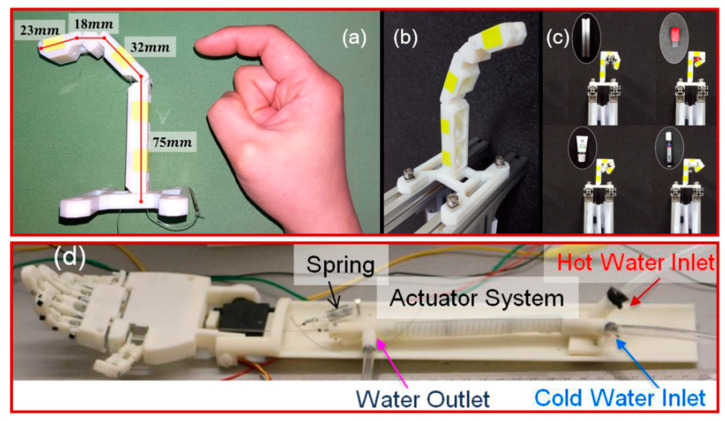
Tendon-driven biomimetic robotic finger: (**a**,**b**) the developed robotic finger and a human finger, and (**c**) grasping various objects (a cosmetic tube, a marker pen, a part of the aluminum profile and a USB flash drive, from left to right) (Reprinted (adapted) with permission [86]. Copyright (2016) SPIE. Digital Library). (**d**) Finger configuration in a prototype hand in the actuation system housed in the forearm and tendons connected to the fingers (Reprinted (adapted) with permission [87]. Copyright (2015) SPIE. Digital Library).

**Figure 5 molecules-25-04246-f005:**
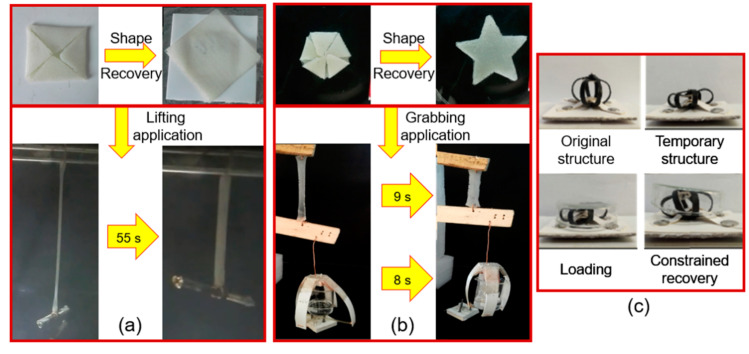
(**a**) Thermally actuated shape-memory behavior and applications of the PUPCL copolymer (Reprinted (adapted) with permission [4]. Copyright (2018) John Wiley and Sons); (**b**) the actuation and grasping behavior of the PUPCL-DS copolymer (Reprinted from Materials and Design [88]. Copyright (2019) Elsevier); (**c**) the deployable device structure made up of cEVA/CF composites shape recovery behavior (Reprinted (adapted) with permission [89]. Copyright (2018) Elsevier).

**Figure 6 molecules-25-04246-f006:**
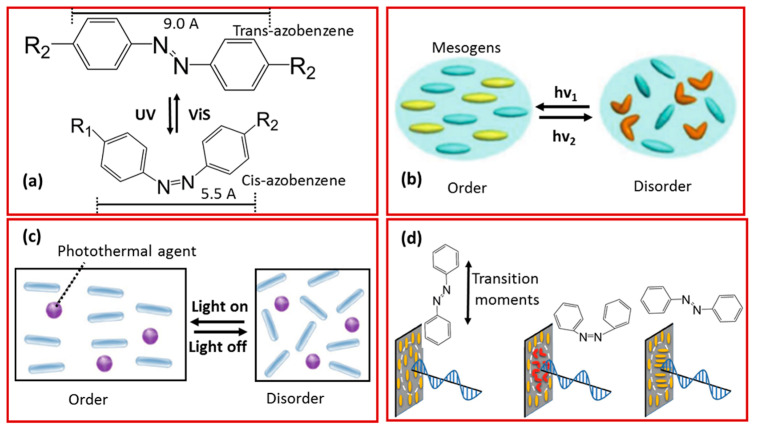
(**a**) Reversible trans–cis photo-isomerization of azobenzene and schematic illustration of (**b**) photo-induced and (**c**) photothermal-induced order–disorder phase transition in LCPs (Reprinted (adapted) with permission [101]. Copyright (2019) John Wiley and Sons). (**d**) Photo-reorientation of azobenzene containing LCPs with linearly unpolarised light (Reprinted (adapted) with permission [96]. Copyright (2019) John Wiley and Sons).

**Figure 7 molecules-25-04246-f007:**
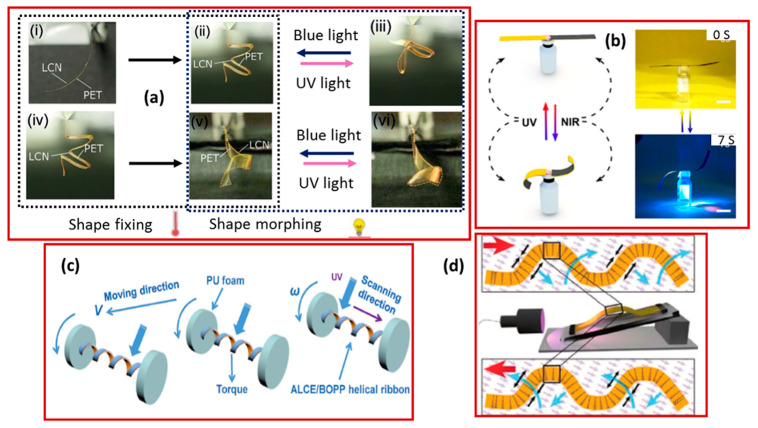
(**a**) Demonstration of shape fixing and shape morphing possibilities using a single actuator strip (Reprinted from Annalen der Physic [105]. Copyright (2012) John Wiley and Sons). (**b**) Schematic diagram and real deformation of the bending state and construction of the actuator under NIR and UV lamp (Reprinted (adapted) with permission [107]. Copyright (2020) American Chemical Society). (**c**) Mechanism of various UV-induced actuators based on bilayer LCPs (Reprinted (adapted) with permission [108]. Copyright (2017) John Wiley and Sons). (**d**) Schematic diagram of a polymeric film that was constrained at both ends and reacted with an oblique-incidence light source (left). The blue arrows show the way the film deformed while the red ones indicate the propagation direction of the wave (Reprinted (adapted) with permission [96]. Copyright (2019) John Wiley and Sons).

**Figure 8 molecules-25-04246-f008:**
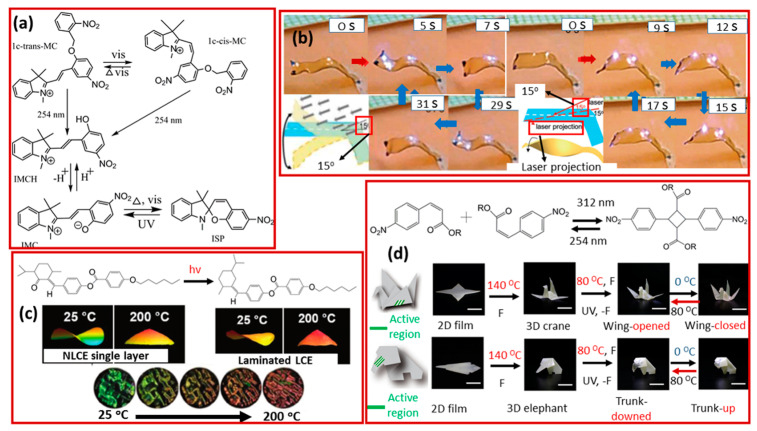
(**a**) Spiropyran-based molecular switch and the UV-vis absorption spectra in response to different stimuli (Reprinted (adapted) from https://pubs.acs.org/doi/10.1021/jacs.8b09523). (**b**) Chemical structure of the liquid crystal elastomer containing 2-butyloctyl carbon chains and autonomous arm-like motions of the strip actuator under constant illumination of a laser (Reprinted (adapted) with permission [110]. Copyright (2012) John Wiley and Sons). (**c**) The photo-isomerization of the chiral olefin and the deformation and color change of the LCE layer when heated from 25 to 200 °C (Reprinted (adapted) with permission [111]. Copyright (2019) John Wiley and Sons) [70]. (**d**) Photo-reversible dimerization of nitro-cinnamate and typical model to illustrate the light programming (Reprinted (adapted) from Science Advances, Applied Sciences and Engineering [112]. Copyright (2018), American Association for the advancement of Science).

**Figure 9 molecules-25-04246-f009:**
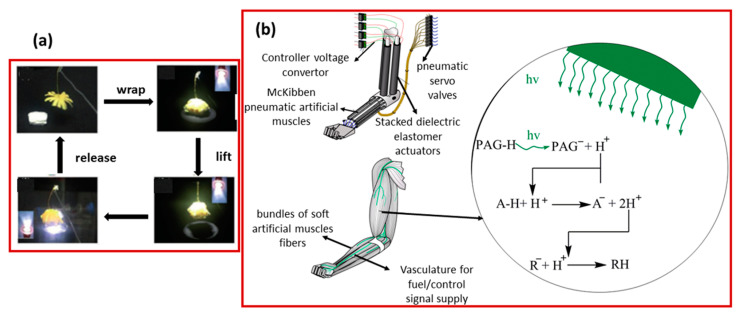
(**a**) BA2DA-PVDF robot with multiple arms grasping objects (Reprinted (adapted) with permission [113]. Copyright (2018) John Wiley and Sons); (**b**) Proposed molecular level-controlled robotic system and detail of the amplifying system copolymer (Reprinted from Scientific Reports [114]. Copyright (2017) Springer Nature).

**Figure 10 molecules-25-04246-f010:**
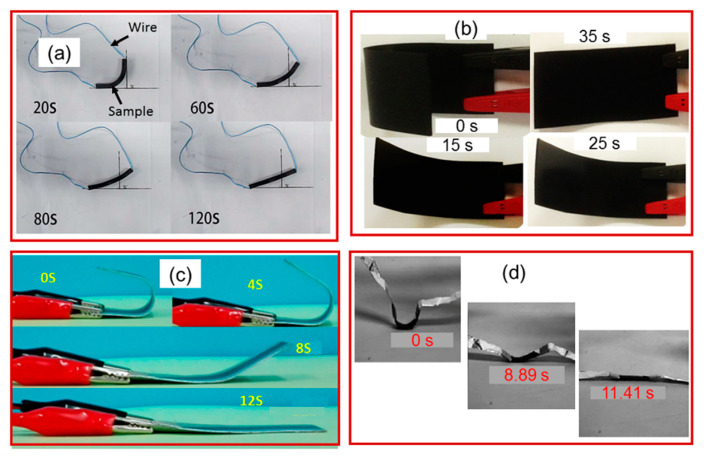
(**a**) The shape-recovery process of the compound aerogel (with a carbon nanotube:graphene weight ratio of 3:5)/epoxy resin composite under a voltage of 60 V (Reprinted from Journal of Materials Chemistry A [128]. Copyright (2015) Royal Society of Chemistry); (**b**) photographs showing the shape recovery process of the representative E-CNT sample obtained at a triggering voltage of 30 V (Reprinted from Materials [130]. Copyright (2015) MDPI); (**c**) demonstration of electro-active shape-memory behavior of PPLACNT-S (Reprinted (adapted) with permission [106]. Copyright (2016) American Chemical Society); (**d**) electroactive shape recovery behavior of the Cu-CNT dispersed PLA/ESO nanocomposite (Cu-CNTs content of 2 wt% at a DC voltage of 40 V) (Reprinted (adapted) with permission [129]. Copyright (2016) John Wiley and Sons.

**Figure 11 molecules-25-04246-f011:**
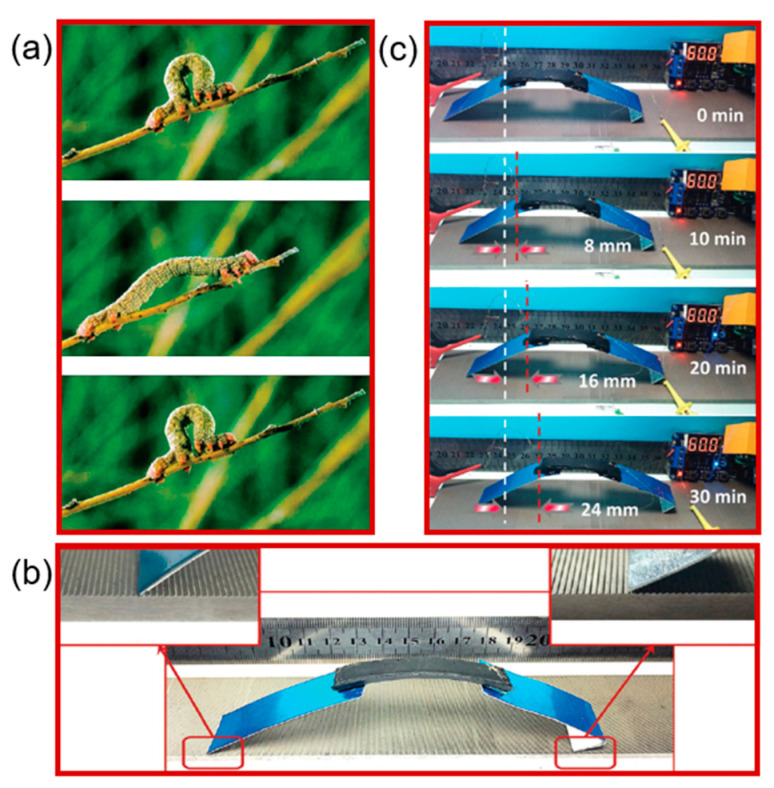
Application of an electro-active shape-memory polymer composite to an inchworm robot. (**a**) Photographs of inchworm locomotion. (**b**) Illustration of a designed inchworm robot with a double layer CNT–SMP composite in connection with two metal plates as moving legs. (**c**) Snapshots of an inchworm-type robot in locomotion. It has moved over 24 mm in 30 min (Reprinted (adapted) with permission [138]. Copyright (2016) Royal Society of Chemistry).

**Figure 12 molecules-25-04246-f012:**
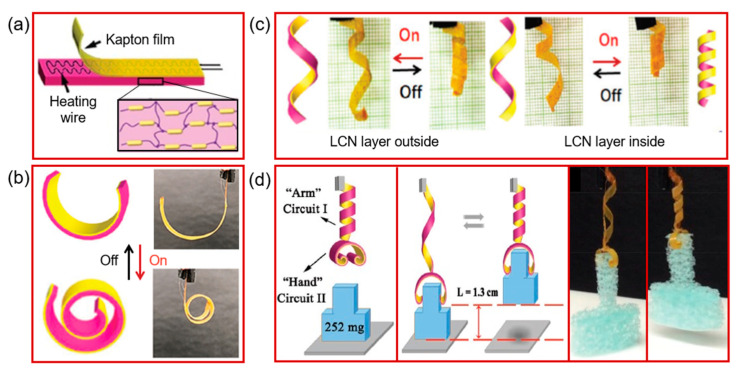
Electro-thermal liquid crystalline network (ELCN) actuators. (**a**) The integrated ELCN actuator; (**b**) the bending states of the ELCN actuator driven by the external voltage; (**c**) illustration of an ELCN actuator strip programmed to have a helical LCN shape with either an outside layer or inside layer, and the corresponding reversible actuation when the power is off and on. (**d**) Photographs of the self-locking gripper made with the ELCN actuator, which can grip and lift an object by operating the programmable hand and arm by using two electric circuits (Reprinted (adapted) with permission [141]. Copyright (2019) John Wiley and Sons).

**Figure 13 molecules-25-04246-f013:**
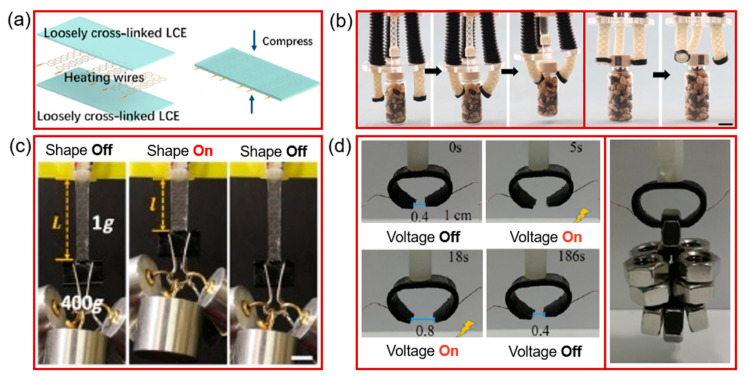
(**a**) Fabrication steps of an LCE-based tubular actuator with three serpentine heating wires sandwiched between two layers of loosely cross-linked LCE film. (**b**) An LCE artificial muscle film that could lift a load of 3.92 N by 38% of its initial length. (**c**) Schematic of the soft gripper and grasping and lifting of the vial (50 g) and twisting the cap of a vial (Reprinted (adapted) from Science Advances, Applied Sciences and Engineering [142]. Copyright (2019), American Association for the advancement of Science). (**d**) Schematic of the PEO/S-CNT composites used to grasp and release an object with the voltage on (36 dc) and off with a picture of the gripper for grabbing nuts (Reprinted (adapted) with permission from Xu et al. [124]. Copyright (2019) American Chemical Society).

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
