# Peer review of "Shape-Memory Polymeric Artificial Muscles: Mechanisms, Applications and Challenges"

_molecules, 2020, doi:10.3390/molecules25184246_

Round 1

Reviewer 1 Report

1.  Very well presented and useful review.

2.  Authors could have covered some more fundamental concepts including constitutive equations (which they mention) for the shape memory effects.

3. The review is nicely classified on the basis of the three factors which induce actuation of the SMPs including thermal, photo and electric.  It may be well worth including some work on ionic EAPs to enhance the review.

4.  Why is there no mention of polypyrrole? It appears to have good potential for Artificial Muscles.

5.  It is very difficult to decipher the details of the figures while reading the text. The figures are too small and lack clarity.

6. Please check in detail for minor typos and grammar.

Author Response

Reviewer-1

Question 1. Very well presented and useful review.

Answer. Thank you for the appreciations, we will try to continue our struggles to produce much better research outcomes.

Question 2. Authors could have covered some more fundamental concepts including constitutive equations (which they mention) for the shape memory effects.

Answer. Thank you for the suggestions. Our main objective is to explain the shape memory polymer artificial muscles. Applying a deeper analysis will make the manuscript more basic.

Question 3. The review is nicely classified on the basis of the three factors which induce actuation of the SMPs including thermal, photo and electric. It may be well worth including some work on ionic EAPs to enhance the review.

Answer. Thank you for the suggestions. Yes, we can try to add ionic electro-active artificial muscle polymers. But we did not add because it is a separate topic and includes various subclasses like ionic polymer gels, ionic polymer metallic composite, electrorheological fluids, and conductive polymers. Further, our review topic is to shape memory polymer-based artificial muscles. We think if we include the ionic EAP, the manuscript will lose its objective. Following are the references which is providing proof to the above context.

  • Mirfakhrai, T.; Madden, J.D.; Baughman, R.H. Polymer artificial muscles. Materials today 2007, 10, 30-38.
  • Bar-Cohen, Y. Current and future developments in artificial muscles using electroactive polymers. Expert Review of Medical Devices 2005, 2, 731-740.
  • Melling, D.; Martinez, J.G.; Jager, E.W. Conjugated polymer actuators and devices: progress and opportunities. Advanced Materials 2019, 31, 1808210.

Question 4. Why is there no mention of polypyrrole? It appears to have good potential for Artificial Muscles.

Answer. Yes, we agree that PPy is a conductive polymer. The activation produced by PPy based polymers/copolymers or composite is not due to shape memory phenomenon. But instead by electrochemical process, i.e., oxidation-reduction: a purely electrochemical actuation. Therefore, if we add, it will lead us to another big topic (electro-active polymers in a broad sense). This way, our research topic will deviate from the main title of our manuscript. The references that help us to limit the context in the above text are as follows:

  • Harjo, M.; Järvekülg, M.; Tamm, T.; Otero, T.F.; Kiefer, R. Concept of an artificial muscle design on polypyrrole nanofiber scaffolds. PloS one 2020, 15, e0232851.
  • Ismail, Y.A.; Shabeeba, A.; Sidheekha, M.P.; Rajan, L. Conducting Polymer/Hydrogel Systems as Soft Actuators. Actuators: Fundamentals, Principles, Materials and Applications 2020, 211-252.
  • Ebadi, S.V.; Fashandi, H.; Semnani, D.; Rezaei, B.; Fakhrali, A. Electroactive actuator based on polyurethane nanofibers coated with polypyrrole through electrochemical polymerization: a competent method for developing artificial muscles. Smart Materials and Structures 2020, 29, 045008.
  • Beregoi, M.; Evanghelidis, A.; Diculescu, V.C.; Iovu, H.; Enculescu, I. Polypyrrole actuator based on electrospun microribbons. ACS applied materials & interfaces 2017, 9, 38068-38075.
  • Mahloniya, R. POLYPYRROLE BASED NANOCOMPOSITES AS RADIATION SHIELDING MATERIALS: A MINI REVIEW.

Question 5. It is very difficult to decipher the details of the figures while reading the text. The figures are too small and lack clarity.

Answer. Thank you. We tried our best to improve the quality and clarity of the Figures, by re-drawing while keeping in mind the copy-right instruction/limits, as shown in the revised manuscript.

Question 6. Please check in detail for minor typos and grammar.

Answer. Thank you, we noted the instructions and tried to remove the typos and grammatical errors, such as ‘l22, l49, l66, l270, l310, l466, l544, l550, l598, l646, l693, l733’ in the revised manuscript.

Reviewer 2 Report

This paper presents a thorough Review on Shape Memory polymeric artificial muscles, with special emphasis in their mechanisms, applications and challenges.

This is a very interesting review on these smart materials: It is well presented, structured and written. The content of this study is interesting, and this kind of reviews are useful for the researchers in the related field. So, in my opinion, it can be accepted after some minor points are addressed:

In general, all the Figures are too small and the explanations inside are difficult to read. I suggest enlarging the pictures or increasing the resolution.

Some sentences in the paper should be revised, in English or in sense. For example:

Line 303 “…the most commonly used representative group…” what does it mean?

Line 357  “They used several forces” which ones?

Line 360 “Consequently, various photo simulation methods and explained the mechanism of the generation of the wave”. This sentence is difficult to understand.

Line 119 and Line 128. Some references should be added to complete the review as, for example, references related to core-shell composites and LCEs: Smart Mater. Struct. 27 (2018) 075013 or J. Phys. Chem. C (2017), 121, 22403−22414; J. Phys. Chem. C (2016) 120 24417–26; J. Polym. Sci. Part B Polym. Phys. (2016), 1, 1–16. Or references related to emerging techniques like 3D printing used as a solution for achieving complex spatial designs: Adv. Mater. (2016) 28 4449–54.

Author Response

Reply to reviewer’s comments

Reviewer-2

Question 1. This paper presents a thorough Review on Shape Memory polymeric artificial muscles, with special emphasis in their mechanisms, applications and challenges.

Answer. Dear reviewer, thank you. We agree that our manuscript is mainly focusing on Shape memory polymer artificial muscles.

Question 2. This is a very interesting review on these smart materials: It is well presented, structured and written. The content of this study is interesting, and this kind of reviews are useful for the researchers in the related field. So, in my opinion, it can be accepted after some minor points are addressed:

Answer. Thank you for your appreciation. We will try our best to improve our knowledge and add better information to the existing list of SMPs research studies.

Question 3. In general, all the Figures are too small and the explanations inside are difficult to read. I suggest enlarging the pictures or increasing the resolution.

Answer. Thank you, we agree with the observations. We tried our best to re-draw the Figures with increased resolution, as shown in the revised manuscript.

Question 4. Some sentences in the paper should be revised, in English or in sense. For example:

àLine 303 “…the most commonly used representative group…” what does it mean?

àLine 357 “They used several forces” which ones?

àLine 360 “Consequently, various photo simulation methods and explained the mechanism of the generation of the wave”. This sentence is difficult to understand.

Answer. Thank you. We made appropriate changes at the mentioned points. Now the new sentences can be read as:

à"The most commonly used representative group", has been revised to "the most commonly used representative materials."

à“They used several forces”, is deleted, a typing error.

à“Consequently, various photo simulation methods and explained the mechanism of the generation of the wave”. This sentence is shifted above and combined with the most commonly used representative materials. Now the new sentence can be read as "They used several photo simulation methods and explained the mechanism of the generation of the wave (stimuli light wave)". I hope now the sense of these sentences will be clear.

Question 5. Line 119 and Line 128. Some references should be added to complete the review as, for example, references related to core-shell composites and LCEs: Smart Mater. Struct. 27 (2018) 075013 or J. Phys. Chem. C (2017), 121, 22403−22414; J. Phys. Chem. C (2016) 120 24417–26; J. Polym. Sci. Part B Polym. Phys. (2016), 1, 1–16. Or references related to emerging techniques like 3D printing used as a solution for achieving complex spatial designs: Adv. Mater. (2016) 28 4449–54.

Answer. At line 119 the above mentioned references are added in form a new sentence. It can be read as: “Along with these other methods were introduced to obtain free-standing [34], autonomous [35], controlled shape memory actuation [36] by using a glassy thermoset-stretched liquid crystalline network, epoxy-based shape memory lightly cross-linked network, and carbon nanotube/epoxy shape memory LCE, respectively”.

  • Belmonte, A.; Lama, G.C.; Cerruti, P.; Ambrogi, V.; Fernández-Francos, X.; De la Flor, S. Motion control in free-standing shape-memory actuators. Smart Materials and Structures 2018, 27, 075013
  • Belmonte, A.; Lama, G.C.; Gentile, G.; Fernández-Francos, X.; De la Flor, S.; Cerruti, P.; Ambrogi, V. Synthesis and characterization of liquid-crystalline networks: toward autonomous shape-memory actuation. The Journal of Physical Chemistry C 2017, 121, 22403-22414.
  • Lama, G.C.; Cerruti, P.; Lavorgna, M.; Carfagna, C.; Ambrogi, V.; Gentile, G. Controlled actuation of a carbon nanotube/epoxy shape-memory liquid crystalline elastomer. The Journal of Physical Chemistry C 2016, 120, 24417-24426.
  • Polym. Sci. Part B Polym. Phys. (2016), 1, 1–16 (we cannot find this reference.)

However, at line 128, the newly added paragraph can be read as:

Additive manufacturing is also gaining popularity in various scientific disciplines for device fabrication, tissue engineering [60]. The 3D-printer extrude the thermoplastic or LCE molten polymer that cools and solidifies to form a 3D structure, when cycled above and below their transition temperature or nematic-to-isotropic transition temperature (TN1). However, their development is limited to the 3D printable functional materials [61]. The thermoplastic/LCE ink with the highest triggerable dynamic bonds can lock controlled network configurations in the form of a 3D shape on exposure to UV light without an imposed mechanical field [60, 62].

The 3D printed reversible shape-changing soft actuators show 2W-shape-changing behavior. The printed conductive wires actuate LCE strip/SMP strips via Joule heating (UV light or heat treatment. The uniaxial deformation of the SMP strip/LCE strip acts as a driving force to achieve bending) [63]. The 3D-printing shapes can be applied to flexible electronic devices, i.e., soft crawler, sensors, self-deploying devices, and implantable medical devices [63-66]. The 3D-printing techniques are now evolving towards 4D-printing and have attracted increasing interests since its development. The materials used in 4D-printing include hydrogels, multi-materials shape-memory composites and LCE. Unlike typical SMPs, the 4D-printing materials can show a triple shape memory effect. These triple shape memory polymers possess two distinct temporary networks, which is allowing them to memorize an additional temporary shape [6, 65, 67]. The fundamental principle of 4D printing is to directly combine the structural design of shape change to the material components and 3D printing processes. It can simplify the design strategy, fabrication process, and realize desirable 4D properties. Compared to the traditional manufacturing process, such as molding and cutting, the 4D-printing process can significantly save fabrication costs. From all this discussion it is clear that the field of artificial muscles is strongly developing and more and more techniques are advancing the field of artificial muscles.

Reviewer 3 Report

The review ’Shape Memory Polymeric Artificial Muscles: Mechanisms, Applications and Challenges’ by Chen et al. gives an application-oriented overview of shape-memory polymers.

SMPs have been heavily researched since the advent of artificial muscles, and some research is still ongoing. It appears that SMPs achieved their peak popularity about 10 years ago, evidenced by a reputable number of review papers from that time (see e.g. https://doi.org/10.1016/j.polymer.2013.02.023). This said, it is surprising to read from the Abstract on that “no proper survey has gone through shape memory artificial muscles” – this is demonstrably not so. Instead, the authors should highlight the most recent advances that are possibly not covered in the previous reviews.

The review suffers in a poor organization.  E.g., the main chapter “3. Applications of …” actually contains a convolute of different topics from the working principle to fabrication techniques, with only occasional mention of applications. After reading, the applications for SMPs were still unclear.

The first expression of the manuscript hints to low professional quality. In the Abstract, it is claimed that the application for the actuators is actuators – an apparent tautology. An actuator is never an application – but actuators can be used in different applications.

The advantages of polymers (line 46-48) are not convincing – I do not think Ni-Ti has too difficult preparation, high price, etc.

The working principles of SMPs in Figure 1 and the corresponding text are confusing regarding the ‘memory’ effect. How does the ‘two-way shape memory effect’ differ from the trivial expression of the thermal expansion coefficient (also relevant to cooling and melting)? The asymmetrical, aligned structures are visible in Fig. 1 bottom row appear to be the key to understanding the concept, also well described in the previous reviews; however, it is not mentioned in this review. Thus, it is unclear whether the Authors have understood it in high detail. For example, the reference [65] (6 years old, so not that recent, although claimed so in the manuscript) was not previously considered being an SMP, but just a means of engaging the huge thermal expansion coefficient of nylon, without shape-memory properties expressed explicitly. The authors are kindly requested to express the working principles in a more consistent manner (also revising the main definition at line 81).

There is frequent confusion between ‘electrically actuated’ (or electric-field-actuated, line 128) and ‘electrothermally actuated’. The majority of the mechanisms claimed as electrically actuated actually use Joule heating (capitalized!), thus fall into the category of electrothermal activation. I’m not sure if I found any example that is electrically activated… As activation voltages for electrothermally responsive actuators are emphasized frequently, it is worth noting that this voltage value is only to satisfy the Ohm’s Law, as the electric field density itself does not induce actuation.

In the Conclusion chapter, ‘How to effectively dissipate the heat when the SMPs are restored to the original shape’ appears to refer to thermally activated SMPs only.

I did not understand why do ionic artificial muscles suffer in high energies (l68)?

What is a “sizeable driven force” (l216)?

The strain is not measured in mm (line 481).

What are ‘electrically cross-linked soft robots (l529)?

The manuscript appears not have passed a proofreading stage, as it contains unfinished sentences and a lot of typos that are easily caught using speller software (e.g., ‘polymeric based’ (l64); black-ink (l163); the ultraviolet (l358); ‘can makes’ (l413);  ‘thermos-active’ (l432)).

Author Response

Reply to reviewer’s comments

Reviewer-3

Question 1. The review ’Shape Memory Polymeric Artificial Muscles: Mechanisms, Applications and Challenges’ by Chen et al. gives an application-oriented overview of shape-memory polymers.

Answer. Thank you. Yes we agree that our manuscript is mainly focusing on the SMP artificial muscles, mechanisms, applications and challenges.

Question 2. SMPs have been heavily researched since the advent of artificial muscles, and some research is still ongoing. It appears that SMPs achieved their peak popularity about 10 years ago, evidenced by a reputable number of review papers from that time (see e.g.https://doi.org/10.1016/j.polymer.2013.02.023). This said, it is surprising to read from the Abstract on that “no proper survey has gone through shape memory artificial muscles” – this is demonstrably not so. Instead, the authors should highlight the most recent advances that are possibly not covered in the previous reviews.

Answer. Yes, we agree that SMPs achieved great popularity in the last 10 years. However, our main objective is to submit an overview on the most recently published SMPs based artificial muscles. Dear reviewer, we read the mentioned literature, we agree that the sentence- "no proper survey has gone through shape memory artificial muscles", is not suiting the abstract exactly. Therefore we changed it into another sentence and it can be read as "This paper presents a review on the recent progress in shape memory polymers based artificial muscles". 

Question 3. The review suffers in a poor organization.  E.g., the main chapter “3. Applications of …” actually contains a convolute of different topics from the working principle to fabrication techniques, with only occasional mention of applications. After reading, the applications for SMPs were still unclear.

Answer. Dear reviewer, thank you for the suggestions. We reviewed the electro-active shape memory artificial muscles part. Throughout the manuscript, in the application part, we tried to give an overview on the shape memory polymers (in each part). After that, we tried to cite the applications of each of these types. Yes, we agree that mostly the electro-active shape memory polymers are very rarely applied to the artificial muscles. However, we tried our best to find some existing literature between 2020-2015. In this regard we tried to add another literature which can be read as:

Similarly, Rubaiai et al.132, prepared self-pneumatic actuator (SPA) by using the shape fixation and shape recovery behavior of conductive poly-lactic acid (CPLA). The fabrication of the SPA was such that the CPLA was integrated into the flat side of the SPA, as shown in Figure 1. The components of the SPA were fabricated using casting and 3D printing. However, local indentations i.e., join 1, 2, 3, were designed in the flat sheet geometry to facilitate bending at the hinge locations. For Joule heating silver wires were soldered to each hinge without affecting the device flexibility. After this the CPLA+silver wire was encapsulated through a silicone rubber bath to allow adhesion with the SPA. When Joule heating is activated at a given location on the CPLA, the material nearby softens and thus enables bending at the location. Using this concept, they prepared actuators which enabled them to grasp multiple objects using different grasping modes. The maximum load hold by SPA was 800 g when the SPA was actuated with 22 psi and all CPLA joints actuated with a 12 V inputs.

Question 4. The first expression of the manuscript hints to low professional quality. In the Abstract, it is claimed that the application for the actuators is actuators – an apparent tautology. An actuator is never an application – but actuators can be used in different applications.

Answer. Thank you for your suggestion. Yes, we agree that the actuator can be used in different applications. Therefore, we changed the sentence and now the new sentence can be read as "The prepared SMP artificial muscles can be used, in a wide range of applications, from biomimetic and soft robotics to actuators, because they can be operated without sophisticated linkage design and can easily be deployed to achieve complex final shapes".

Question 5. The advantages of polymers (line 46-48) are not convincing – I do not think Ni-Ti has too difficult preparation, high price, etc.

Answer. The references from which the sentence i.e., line 46-48 is taken are included here. 

  • Ratna, D.; Karger-Kocsis, J. Recent advances in shape memory polymers and composites: a review. Journal of Materials Science 2008, 43, 254-269.
  • Sahoo, N.G.; Jung, Y.C.; Goo, N.S.; Cho, J.W. Conducting shape memory polyurethane‐Polypyrrole composites for an electroactive actuator. Macromolecular Materials and Engineering 2005, 290, 1049-1055.
  • Sahoo, N.G.; Jung, Y.C.; Yoo, H.J.; Cho, J.W. Influence of carbon nanotubes and polypyrrole on the thermal, mechanical and electroactive shape-memory properties of polyurethane nanocomposites. Composites Science and Technology 2007, 67, 1920-1929.
  • Meng, H.; Li, G. A review of stimuli-responsive shape memory polymer composites. Polymer 2013, 54, 2199-2221.

Question 6. The working principles of SMPs in Figure 1 and the corresponding text are confusing regarding the ‘memory’ effect. How does the ‘two-way shape memory effect’ differ from the trivial expression of the thermal expansion coefficient (also relevant to cooling and melting)? The asymmetrical, aligned structures are visible in Fig. 1 bottom row appear to be the key to understanding the concept, also well described in the previous reviews; however, it is not mentioned in this review. Thus, it is unclear whether the Authors have understood it in high detail. For example, the reference [65] (6 years old, so not that recent, although claimed so in the manuscript) was not previously considered being an SMP, but just a means of engaging the huge thermal expansion coefficient of nylon, without shape-memory properties expressed explicitly. The authors are kindly requested to express the working principles in a more consistent manner (also revising the main definition at line 81).

Answer. Dear reviewer, thank you for the suggestion. We tried our best to revise this part. Now it can be read as follows:

Artificial muscle, a generic term used for a class of bio-inspired materials and devices that can reversibly expand, contract or rotate within one component due to an external stimulus (such as voltage, current, temperature, and light, etc.) These three actuation responses can be combined within a single component to produce other types of motions (e.g., bending, contraction on one side of the material, and expansion on the other side).[25, 26] Various techniques were used to produce artificial muscles in the past. For example, rubber was used for pneumatic artificial muscles, in which a gas (for example, air) was used as the energetic source to control or expand the rubber bladder. For conductive polymer materials, electric-current or voltage was used as the deriving energy source.[25-27] However, for the actuation of shape-memory materials, an external stimulus (e.g., light, heat, or voltage, etc.) was used as a source of actuation. The artificial muscles based on these materials were termed as pneumatic/electro-active/shape memory artificial muscles. All of these SMPs and artificial muscles were observed with specific limits. Deeper insight into the working mechanism of polymeric shape memory polymers and artificial muscles is as follows:

2.1. Shape Memory Effects

SMPs are stimuli-responsive smart materials that can undergo recoverable deformations upon application of an external stimulus. This phenomenon in SMP stems from a dual segment system, i.e., the cross-links that determines the permanent shape and the switching segments with a transition temperature (Ttrans) that fix the temporary shape. Below Ttrans the SMPs remain stiff, while they become relatively soft upon heating above Ttrans. Consequently, they can be deformed into a desired temporary-shape upon applying an external force. While cooling and then subsequently removing this external force their temporary-shape can be maintained for a long time. However, upon re-heating their temporary deformed shape will automatically recover the original permanent shape. From here, it is clear that each SMP consists of dual-segments, one is highly elastic, and it can be molecular entanglement, crystalline phase and chemical cross-linking, etc. The other is a reversible domain that determines the temporary shape and reduces its stiffness upon a particular stimulus. It is usually related to crystallization/melting transition, glass transition, and reversible molecular cross-linking structure, etc. Upon triggering the strain energy stored in the temporary shape is released, which results in shape recovery. These fixed and reversible domain determines the shape memory performance of SMPs [8, 23, 28].

The SMPs can be divided into those showing one-way and two-way shape-memory effects (SME) as shown in Figure 1. The one-way SME is irreversible, meaning that once the shape-memory process terminates, the SMP is fixed to a specific structure artificially to restart the shape memory process. Such type of SMPs performance is termed as one-way shape memory effect (SME). In these SMPs the transitions from temporary shape to permanent shape cannot be repeated by simply reversing the stimulus. Here the shape changing occurs only in one direction (Figure 1a). Every time a new programming process is necessary to achieve the temporary shape (after recovery process) [29, 30] . These can be divided into dual-SMPs and multi-SMPs. If the SMPs remember only on temporary shape, then these SMPs are termed as dual-SMPs materials. However, if the SMPs remember two or more temporary shapes then they are termed as triple or multiple-SMPs. In multiple-SMPs the deformed material return to their original shape step-by-step from two or more temporary shapes, as shown in Figure 1b.

Obvious realization of multi-shape memory effect (multi-SME) is mainly determined by two kind of strategies. One strategy is to use polymers with a broad thermal transition in which multiple thermal transitions and temporary shapes are programmed at multiple temperatures across the broad transitions (with different composition materials). In this system, mostly a blend is prepared with broad glass transition that varies with the blend composition (due to miscibility). The other method to achieve broad thermal transition is to graft, block copolymerization of different components or the chemical crosslinking coupled with supramolecular bonding etc. However, the important point to note is that, it is very difficult to obtain broad thermal transition with chemical reaction (due to its complex nature). Further, the method based on miscible polymer blends are limited because most polymer blends are immiscible. This is why, very less research interests has been dedicated to these types of SMPs [28, 31]. The other kind of strategy to achieve multiple-SME is to construct several domains with well separated thermal transitions. This method is obtained by blending two chemically crosslinked polymers, copolymers or composites. In these blends the reversible domains are related to the two crystallization/melting transitions, or one crystallization/melting temperature and one glass transition temperature of the polymers/composites [29]. This strategy to achieve multi-SME is more exciting because SME is endowed by controlling appropriate microstructure.

However, the two-way SME responds entirely to external stimuli and is reversible and does not require additional programming of the material itself. Liquid crystal elastomer [32] , cross-linked crystalline polymer [33]  and their composites show these features [34] . The two-way reversible shape-memory effect can further be subdivided into quasi two-way and stress-free two-way shape memory effects [35] . The quasi-2W-SME can be observed both in LCEs and semi-crystalline networks under an external stress. Since LCEs are elastic polymer networks containing main chain or side-chain liquid crystal units (LC-units). These LC-units are capable of undergoing reversible mesomorphic-to-isotropic phase transitions. In a liquid crystal these domains are typically randomly oriented with respect to each other, thus these are called liquid crystalline polydomains. During confirmation of LCE network, these polydomains can be aligned when external filed is applied (e.g., magnetic field or stretching force). This results in the alignment of the monodomains elongations of LCE strip. When heated to a temperature above liquid crystal clearing temperature (Tcl), the polymer chains world reduce its anisotropy. Hence macroscopic contraction of the sample occurs upon cooling below Tcl , the sample would revert to the original anisotropic state (elongation). This process is fully reversible and the monodomains can be formed either physically or by chemical process by using two step crosslinking process or one step crosslinking process [29, 36, 37] . During two step crosslinking process, in the 1st step isotropic network is established via partial crosslinking then anisotropy is induced (via deformation) and fixed further by crosslinking in the second step [38-41] . However, in the one step crosslinking small molecules or polymeric liquid crystal precursors are macroscopically oriented by applying an external field. After that aligned precursors are polymerized/crosslinked to form a macroscopically anisotropic LCE [42-45] . Where in physical process the monodomains formation occurs via hanging an external weight or stress to the already synthesized LCE polydomains [46] . The basic difference between these two crosslinking methods is that the chemically crosslinked method cannot be alter after fabrication process, while the physical fabricated network can be tuned easily (by applying external stress). This quasi-2W-SME can also be observed in semi-crystalline networks under constant tensile load. The semi-crystalline network (of polycyclooctene) underwent elongation when they are cooled across the Tm (i.e., crystallization induced elongation or CIE). When heated above the Tm under same load, the elongation reversed (i.e., melting induced contraction or MIC) [42] . The CIE-MIC transformation for the semi-crystalline networks require the presence of an external force. Besides this, the crosslink density is considered a tailoring parameter to control the quasi two-way shape memory response [47-50] . If we look back at the mechanism of LCE, the anisotropic alignment of polymer chain is the true inherent mechanism for semi-crystalline polymer network. Although, it is the external stress that is turning the anisotropy and strain change. But, the requirement of an external stress is serious limitation for the potential devices application of quasi-2W-SMPs. Therefore the search for alternative mechanisms and materials to enable stress free 2W-SME has been a constant chases between SMPs community.

In this regard landlein’s group[34] synthesized polyester urethane (PEU) network with poly (ω-pentadecalactone) (PPDL) and PCL segment. The basic steps were similar in mechanism to irreversible multi-SME (triple -SME) (Figure 1b), but no force was required for the cyclic actuation. The two polyester provided a high Tm (Tm,high) around 64 oC and a low Tm (Tm,low) around 34 oC, respectively. Original shape of PEU sample i.e., (Shape S) was first deformed at a Treset>Tm, high by applying an external force. This deformation was fixed by obtaining shape S1, at a lower temperature (Tlow), i.e., Tlow<Tm,low, while removing external force, as shown in Figure 1c. At this moment the chain confirmation associated with PPDL were changed. After this, the PEU sample was reheated again to a temperature Thigh, i.e., (Tm,low<Thigh<Tm,high), leading to another shape, i.e., S2. During this time the anisotropy and chain confirmation of crystalline phase of PCL were changed. Upon reheating to Thigh, the partial orientation in the PCL chain is removed and the deformation fixed by PPDL domain remain untouched. This behavior is setting the network anisotropy for the PCL domain without external force. Hence, macroscopic CIE-to-MIC transformation of PCL domains are induced without external force. Overall, it is this internally created network anisotropy that is reversible and is differentiating reversible 2W-SME from the irreversible multi-SME.

2.2. Programmable Shape-Memory Polymeric Artificial Muscles

---------From l268-l325--------

Question 7. There is frequent confusion between ‘electrically actuated’ (or electric-field-actuated, line 128) and ‘electrothermally actuated’. The majority of the mechanisms claimed as electrically actuated actually use Joule heating (capitalized!), thus fall into the category of electrothermal activation. I’m not sure if I found any example that is electrically activated… As activation voltages for electrothermally responsive actuators are emphasized frequently, it is worth noting that this voltage value is only to satisfy the Ohm’s Law, as the electric field density itself does not induce actuation.

Answer. Yes, we agree that the electro-active and electrically actuated process is the same as an electro-thermal process. Where external potential i.e. voltage is used to provide Joule heating by the same basic concept of Ohm's law. Therefore we replaced all the terms with electro-active (electrically actuated) words throughout the manuscript. This will means electro-thermal actuation where heat is produced via Joule heating to actuate the shape-memory phenomenon.

Question 8. In the Conclusion chapter, ‘How to effectively dissipate the heat when the SMPs are restored to the original shape’ appears to refer to thermally activated SMPs only.

Answer. Thank you. Dear reviewer as you mentioned above (in question No.7) the main concept of SMPs actuation lies within the heating process, although the stimulus remain different. Therefore, where there is the use of heat for the actuation of SMPs it will need dissipation. Therefore, we included this as a challenge in the conclusion part.

Question 9. I did not understand why ionic artificial muscles suffer in high energies (68).

Answer. The ionic polymer-metal composites consist of a solvent swollen ion-exchange polymer membrane. The polymer membrane is laminated by two thin-flexible metal plates (typically percolated pt nanoparticles or Au) or CNTs. The application of a bias voltage to the device causes the migration of mobile ions within the film to an oppositely charged electrode. This movement causes the process of swelling and shrinkage on the opposite sides of the membrane, resulting in bending movement. But if the polymer is electrically conductive as in conducting polymers or CNTs. Then these ions serve to balance the charge generated on these conductors as the potential is changed. This behavior makes a strong local field in the presence of low applied voltage. At this time, high energy is generated due to low distance between ions and electronic charges and the transfer of charges.

  • Mirfakhrai, T.; Madden, J.D.; Baughman, R.H. Polymer artificial muscles. Materials today 2007, 10, 30-38.
  • Brochu, P.; Pei, Q. Advances in Dielectric Elastomers for Actuators and Artificial Muscles. Macromolecular Rapid Communications 2010, 31, 10-36, doi:10.1002/marc.200900425.

Question 10. What is a “sizeable driven force” (216)?

Answer. The sizeable driven force is used in terms of the applied force required for obtaining deformed shapes (shape fixation) and the potential/heat/light intensity required to obtain original shapes after shape fixation. 

Question 11. The strain is not measured in mm (line 481)?

Answer. Corrected.

Question 12. What are ‘electrically cross-linked soft robots (l529)?

Answer. Thank you. Now the new sentence can be read as,….. Prepared soft robots”.

Question 13. The manuscript appears not have passed a proofreading stage, as it contains unfinished sentences and a lot of typos that are easily caught using speller software (e.g., ‘polymeric based’ (l64); black-ink (l163); the ultraviolet (l358); ‘can makes’ (l413);  ‘thermos-active’ (l432)).

Answer. All these and other errors are removed and mentioned at appropriate points.

we are passing special thanks to the reviewers, for their precious time and valuable comments. Thank you once again.

Round 2

Reviewer 3 Report

The new version is much improved and suitable for publication in Molecules.